# Identification and Pharmacokinetic Studies on Complanatuside and Its Major Metabolites in Rats by UHPLC-Q-TOF-MS/MS and LC-MS/MS

**DOI:** 10.3390/molecules24010071

**Published:** 2018-12-25

**Authors:** Yu-Feng Yao, Chao-Zhan Lin, Fang-Le Liu, Run-Jing Zhang, Qiu-Yu Zhang, Tao Huang, Yuan-Sheng Zou, Mei-Qi Wang, Chen-Chen Zhu

**Affiliations:** Institute of Clinical Pharmacology, Guangzhou University of Chinese Medicine, No. 12 Jichang Rd, Guangzhou 510405, China; 20171112685@stu.gzucm.edu.cn (Y.-F.Y.); 20172112126@stu.gzucm.edu.cn (F.-L.L.); 20162112129@stu.gzucm.edu.cn (R.-J.Z.); 20171112687@stu.gzucm.edu.cn (Q.-Y.Z.); 20161112638@stu.gzucm.edu.cn (T.H.); 20161112633@stu.gzucm.edu.cn (Y.-S.Z.); 20172112148@stu.gzucm.edu.cn (M.-Q.W.)

**Keywords:** complanatuside, metabolism, pharmacokinetics, UHPLC-Q-TOF-MS, HPLC-MS/MS

## Abstract

The metabolic and pharmacokinetic studies on complanatuside, a quality marker of a Chinese materia medicatonic, *Semen Astragali Complanati*, were carried out. The UHPLC-Q-TOF/MS (ultra-high performance liquid chromatography coupled with electrospray ionization tandem quadrupole-time-of-flight mass spectrometry) method was applied to identify the metabolites of complanatuside in rat plasma, bile, stool, and urine after oral administration at the dosage of 72 mg/kg. Up to 34 metabolites (parent, 2 metabolites of the parent drug, and 31 metabolites of the degradation products) were observed, including processes of demethylation, hydroxylation, glucuronidation, sulfonation, and dehydration. The results indicated glucuronidation and sulfonation as major metabolic pathways of complanatuside in vivo. Meanwhile, a HPLC-MS method to quantify complanatuside and its two major metabolites—rhamnocitrin 3-*O*-β-glc and rhamnocitrin—in rat plasma for the pharmacokinetic analysis was developed and validated. The T_max_ (time to reach the maximum drug concentration) of the above three compounds were 1 h, 3 h, and 5.3 h, respectively, while the C_max_ (maximum plasma concentrations)were 119.15 ng/mL, 111.64 ng/mL, and 1122.18 ng/mL, and AUC(0-t) (area under the plasma concentration-time curve) was 143.52 µg/L·h, 381.73 µg/L·h, and 6540.14 µg/L·h, accordingly. The pharmacokinetic characteristics of complanatuside and its two metabolites suggested that complanatuside rapidly metabolized in vivo, while its metabolites—rhamnocitrin—was the main existent form in rat plasma after oral administration. The results of intracorporal processes, existing forms, and pharmacokinetic characteristics of complanatuside in rats supported its low bioavailability.

## 1. Introduction

Flavonoids, one of the three most bioactive chemical components (saponins, alkaloids, and flavonoids) in traditional Chinese medicine, are abundant in plants. Thanks to their unique molecular structures, flavonoids have extensive pharmacological effects; for instance, radio-protection [1,2], hepato-protection [3], anti-oxidant [4], anti-hypertension [5], anti-inflammation [6], and anti-aging [7,8] properties. Usually, flavonoids mainly exist as glycosides in plants [9,10]. Previous pharmacokinetic investigations on flavonoids indicate that almost all of them are not absorbed in the small intestines with their prototypes [11], which is consistent with their low bioavailability [12]. Further research on their intracorporal processes reveal that flavonoids have multifarious metabolites in vivo [13], hinting that the metabolites may be the crucial components responsible for the efficacy or safety of their prototypes. Meanwhile, numerous studies have shown that flavonoids and their metabolites, such as flavonoid polyphenols, are capable of scavenging oxygen radicals, and have anti-inflammatory and other biological activities [14,15,16,17]. Therefore, it may be an efficient approach for clarifying the true effective ingredient and their mechanisms by analyzing the metabolites and metabolic behaviors of flavonoids in vivo.

Complanatuside, comprised of a rhamnocitrin and two glucoses at C-3 and C-4′, respectively, exists in *Semen Astragali Complanati*, a commonly used traditional Chinese medicine tonic that originated from the dried seeds of *Astragalus Complanatus* R. Br. [18], and functions as a chemical marker for quality control of the drug [19]. A previous pharmacokinetic study on it [20] shows its T*_max_* (time to reach the maximum drug concentration) at 1.08 h, C_max_ (maximum plasma concentration) of 110.8 ng/mL, and AUC_0-t_ (area under the plasma concentration-time curve) of 566.0 ng h/mL at a dosage of 30 mg/kg after oral administration. The above results suggest that complanatuside metabolized rapidly with lower absorption in rats. Furthermore, rhamnocitrin 3-*O*-β-glc (RNG) and rhamnocitrin (RNC), the degradation products of it, possess evident anti-oxidant [21], anti-inflammatory, and anti-proliferative activities [22]. Thus, complanatuside may act as a pro-drug, and plays pharmacological effects through its metabolites. However, no research on systematic metabolic profiles and pharmacokinetic characteristics of complanatuside and its main metabolites have been reported till now.

Recently, the metabolic profiles based on ultra-high performance liquid chromatography coupled with electrospray ionization tandem quadrupole/time of flight mass spectrometry (UHPLC-Q-TOF/MS and LC-MS/MS), have been demonstrated to be popular approaches for metabolite identification and pharmacokinetic studies of natural products [23,24] with several advantages, such as high sensitivity and good selectivity, and widely applied for the metabolism of flavonoids [25]. As part of our series of studies on bioactivities and pharmacological mechanisms of natural flavonoids, a systemic metabolic profile of complanatuside in rat plasma, bile, stool, and urine was analyzed by UHPLC-Q-TOF/MS. Moreover, a sensitive and rapid LC-MS/MS method was established to simultaneously determine the concentrations of complanatuside together with its two significant metabolites in rat plasma in the present study.

## 2. Results

### 2.1. Metabolites Study

#### 2.1.1. Fragmentation Studies of Complanatuside Standard

Metabolite identification using the UHPLC-Q-TOF-MS/MS method requires a comprehensive understanding of the fragmentation behaviors of the parent compound for reference. In this study, solutions of complanatuside, rhamnocitrin 3-*O*-β-glc, and rhamnocitrin prepared in 50% acetonitrile–water were used for the fragmentation pattern study, which is helpful in metabolite characterizations. Complanatuside (*m*/*z* 623.1682) produced ions at *m*/*z* 461.1089 (C_22_H_21_O_11_) by loss of the Glu moiety, which then produced ions at *m*/*z* 299.0589 (C_16_H_11_O_6_) by loss of the Glu moiety again, as shown in Figure 1A. For rhamnocitrin 3-*O*-β-glc moiety (*m*/*z* 461.1065) as shown in Figure 1B, fragment ions at 299.0568 (C_16_H_11_O_6_) were formed from the loss of the Glu moiety. The rhamnocitrin moiety produced ions at *m*/*z* 271.0594 (C_15_H_11_O_5_) and *m*/*z* 255.0281 (C_15_H_11_O_4_) through the loss of CO and OH, respectively. Similarly, rhamnocitrin (*m*/*z* 299.0560) yielded a product ion at 283.0232 (C_16_H_11_O_5_), which was probably due to the loss of OH moiety, as shown in Figure 1C. Meanwhile, the rhamnocitrin moiety also produced ions at *m*/*z* 271.0606 (C_15_H_11_O_5_) and *m*/*z* 255.0281 (C_15_H_11_O_4_) through the loss of CO and OH, respectively. Besides, the rhamnocitrin also produced ions at *m*/*z* 165.0180 (C_9_H_9_O_3_) and *m*/*z* 151.0018 (C_8_H_7_O_3_) via loss of C_7_H_2_O_3_ and C_8_H_4_O_3_ moiety, respectively.

#### 2.1.2. UHPLC–Q-TOF-MS/MS Analysis

The full scan mass spectrometry of rat plasma, bile, stool, and urine samples before and after oral administration of complanatuside were collected from UHPLC-Q-TOF-MS as shown in Figure 2. By carefully comparing the data of complanatuside-treated samples with those from their corresponding blank samples, 34 metabolites (**M1**–**M34**) for complanatuside in vivo were confirmed as shown in Table 1, and the fragmentation pathways of it were proposed in Figure 3. All the metabolites were detected in plasma, figures are in the Appendix A.

#### 2.1.3. Parent Drug and Metabolites of the Parent Drug

##### Parent compound **M1**

Compound **M1** was detected in rat plasma in a retention time of 0.79 min. **M1** with *m*/*z* 669.1663 (C_28_H_32_O_16_) was 45 Da higher than *m*/*z* 624.1690 with an additional ion +COOH, indicating that **M1** was the ion of the *m*/*z* 624.1690. The fragment ion at *m*/*z* 461.1089 was generated by deglycosylation from the loss of **M1**. Additionally, the fragment ion at *m*/*z* 299.0553 was generated by deglycosylation from the ion at *m*/*z* 461.1089. Based on the above statement, **M1** was identified as complanatuside [20].

##### Metabolites **M2** and **M15**

As shown in Figure 2 and Appendix A, the [M − H]^−^ ion of **M2** was *m*/*z* 461.1089 (C_22_H_22_O_11_), 162 Da lower than *m*/*z* 624.1690. The fragment ion at *m*/*z* 299.0550 (C_16_H_12_O_6_) was generated by deglycosylationfrom the loss of *m*/*z* 461.1089. Similarly, the [M − H]^−^ ion of **M15** was 162 Da lower than the ion of **M2**. Therefore, **M2** and **M15** were identified as the deglucosylated metabolites rhamnocitrin 3-*O*-glucoside and rhamnocitrin [24], respectively.

#### 2.1.4. Metabolites of the Degradation Products

##### Methylated metabolites (**M3**; **M16**)

In rat plasma, **M3** with *m*/*z* 475.1246 (C_23_H_24_O_11_, retention time 3.20 min) and **M16** with *m*/*z* 331.0460 (C_15_H_10_O_6_, retention time 0.92 min) were 14 Da (CH_2_) higher than **M3** and **M15**, indicating that they were methylated metabolites of complanatuside. The fragment ion at *m*/*z* 429.2995 (M−H_2_O -CO) was generated from **M3** and the fragment ion at *m*/*z* 285.0312 (M−CH_2_) was generated from **M16**. Therefore, **M3** and **M16** were identified as methylated metabolites of complanatuside.

##### Demethylated metabolites (**M4**; **M17**)

Two metabolic products **M4** [M − H]^−^ and **M17** [M – H]^–^ at *m*/*z* 447.0929 (C_21_H_20_O_11_) and 331.0460 (C_15_H_10_O_6_) were 14 Da (CH_2_) lower than that of **M2** and **M15**, respectively. This finding suggested that **M4** and **M17** were demethylation products of complanatuside. The fragment ions at *m*/*z* 296.9979, 164.837, and 149.0965 also reviewed the same results.

##### Demethoxy metabolites (**M5**; **M6**; **M7**; **M18**)

Similar to the methylated/demethylated metabolites, the ions of **M5** and **M18** at *m*/*z* 431.0983 (C_21_H_20_O_10_) and 269.0449 (C_15_H_10_O_5_), were likely produced by the loss of methoxy after the metabolites of deglycosylation of complanatuside **M2** and **M15**, respectively. Meanwhile, **M6** and **M7** can also be considered as metabolites of **M3** and **M4** after this transform, respectively.

##### Hydroxylated metabolites (**M8**; **M9**; **M19**; **M20**; **M21**; **M22**)

Several metabolites were generated from complanatuside by hydroxylation, including **M8**, **M9**, **M19**, **M20**, and **M21**, which were 16 Da higher than the metabolites **M4**, **M7**, **M15**, **M16**, and **M17**, respectively. Therefore, they were identified as complanatuside hydroxylated metabolites, and the metabolite **M22** was obtained after hydroxylation of the demethoxy-demethylated metabolites.

##### Sulfonated metabolites (**M10**; **M23**; **M24**; **M25**; **M26**; **M27**; **M28**; **M29**; **M30**)

In rat biological metabolism, sulfonation is a very common metabolic process and its metabolites can be performed simultaneously with other metabolic modalities. We have already detected ninesulfonated metabolites in this experiment, marked as **M10**, **M23**, **M24**, **M25**, **M26**, **M27**, **M28**, **M29**, and **M30**.

##### Glucuronidated metabolites (**M11**; **M12**; **M13**; **M14**; **M31**; **M32**; **M33**)

In addition, the glucuronidated process is a common phenomenon of natural products. Many metabolites were generated after glucuronidation, including **M11**, **M12**, **M13**, **M14**, **M31**, **M32**, and **M33**, identified by secondary mass spectrometry, and some of them have different cleavage patterns due to the addition of glucuronic acid. No glucuronide conjugate of **M1** complanatuside was detected in rat plasma.

##### Dehydrated metabolites (**M34**)

Dehydration is a common metabolic method of flavonoids. Compound **M34** was detected in a retention time of 1.21 min. **M34** with *m/z* 653.1761 (C_28_H_32_O_15_) was 45 Da (with additional ion +COOH) higher than *m*/*z* 608.1741, which showed that **M34** was the ion of the *m*/*z* 608.1741. Meanwhile, the *m*/*z* 608.1741 was 16 Da lower than complanatuside *m*/*z* 624.1690. Metabolite **M34** was identified as the dehydration of complanatuside.

### 2.2. Pharmacokinetic Study

#### 2.2.1. Selection of Internal Standard

To determine the concentration of complanatuside, rhamnocitrin 3-*O*-glucoside, and rhamnocitrin in rat plasma after an oral administration of 72 mg/kg, quercetin was selected as the internal standard (IS) based on the similar chemical structures, chromatographic performance, and ionization under the same conditions.

#### 2.2.2. Method Validation

##### Selectivity and Carryover

The structural formula and MS/MS of blank plasma, as shown in Figure 4A, blank plasma spiked with the internal standard, as shown in Figure 4B, and the real plasma samples, as shown in Figure 4C, at 2 h are depicted in Figure 4. The retention times of complanatuside, rhamnocitrin 3-*O*-β-glc, rhamnocitrin, and IS were 2.9, 3.7, 3.6, and 4.9 min, respectively. There are no significant endogenous substances or metabolite interference was observed at the retention time of them.

##### Linearity and Sensitivity

The good linearity of complanatuside, rhamnocitrin 3-*O*-β-glc, and rhamnocitrin were achieved when the calibration curve was established by the peak area of analytes to IS (Y) versus analyte concentration (X) over the linear concertation rages (*R*^2^ > 0.99) and the lower limit of quantification (LLOQ) of them is 5.20 ng/mL, 2.04 ng/mL, and 1.02 ng/mL, respectively, as shown in Table 2, which were already adequate for the detection in the pharmacokinetic study.

##### Recovery and Matrix Effect

The extraction recoveries of complanatuside, rhamnocitrin 3-*O*-β-glc, and rhamnocitrin were obtained using a protein precipitating method. The average recovery of them at 20, 125, and 50 ng/mL was in the range of 73.5~90.3%. Moreover, the matrix effect at the three quality control (QC) samples were 102.1± 4.4, 92.7 ± 14.6, and 98.7 ± 2.9 ng/mL for complanatuside; 86.0 ± 7.5, 97.7 ± 2.1, and 98.2 ± 2.9 ng/mL for rhamnocitrin 3-*O*-β-glc, and 101.1 ± 4.3, 93.2 ± 2.3, and 95.6 ± 4.8 ng/mL for rhamnocitrin, respectively, as shown in Table 3.

##### Stability

The stability of three compounds in rat plasma (20, 125, and 500 ng/mL) under different conditions were stable, as shown in Table 4, and the relative standard deviations (RSDs) of them were lower than 15%.

##### Precision and Accuracy

The data of precision and accuracy of intra- and inter-day are listed in Table 5. The intra-day and inter-day precision of the three analytes were all in the range 2.7–5.8% and 1.7–10.1%. The accuracy values ranged from −0.9% to 4.0% for intra-day and from −1.2% to 6.5% for inter-day. The results indicated that the method was accurate, reliable, and precise.

#### 2.2.3. Pharmacokinetics of Complanatuside and Two Metabolites

A new LC-MS method was applied in the pharmacokinetic study of complanatuside and its two metabolites after successful oral administration of complanatuside in rats. The concentrations were calculated by the calibration curves, and pharmacokinetic parameters were counted with a non-compartment model according to the concentration-time data used in DAS 3.0 software. The concentration-time curve was depicted in Figure 5, and the pharmacokinetic parameters were shown in Table 6. The T_max_ of complanatuside, rhamnocitrin 3-*O*-β-glc, and rhamnocitrin was 1 h, 3 h, and 5.3 h, respectively, while the T_1/2_ (MRT, mean residence times) were 0.5 h (1.72 h), 1.3 h (4.3 h), and 4.2 h (4.9 h), respectively, which indicated that the speeds of elimination of three analytes were complanatuside > rhamnocitrin 3-*O*-β-glc > rhamnocitrin. Furthermore, for complanatuside, rhamnocitrin 3-*O*-β-glc, and rhamnocitrin, the area under the curve was 143.52 ± 15.73 µg/L·h, 381.73 ± 24.13 µg/L·h, 6540.14 ± 433.70 µg/L·h, and the C_max_ values were 119.15 ± 11.25 ng/mL, 111.64 ± 14.68 ng/mL, and 1122.18 ± 113.32 ng/mL, individually. It manifested that if the complanatuside absorbed as prototype in vivo, then it was rapidly suffered from biotransform as deglycosylation, then rhamnocitrin 3-*O*-β-glc and rhamnocitrin occurred successively. The C_max_ and AUC_0-∞_ of rhamnocitrin were the highest among the three analytes, which indicated that it may be the real potential active ingredient after oral administration by complanatuside in vivo. The descriptions of pharmacokinetic characteristics of complanatuside and its metabolites contributed to clarifying the metabolic process of complanatuside in vivo. Although the bioavailability of complanatuside was low, the biotransform method was helpful to raise its availability.

## 3. Discussion

In this study, a strategy is described using ultra-high performance liquid chromatography quadrupole-time-of-flight mass spectrometry (UHPLC-Q-TOF/MS) with automated data analysis for the rapid analysis of the metabolic profile of complanatuside in rat plasma after oral administration. As a result, a total of 34 metabolites were identified and their characteristic fragmentations were summarized.

Flavonoids are extensively metabolized in vivo by phase II enzymes such as uridine-5-diphosphate glucuronosyltransferases (UGTs) and sulfotransferases (SULTs) to glucuronides and sulfonation metabolites [26]. The efficiency of glucuronidation of flavonoids was very high, followed by sulfonation, with only a very minor contribution by CYP-mediated oxidation. This metabolism rank could be proved by the most similar flavonoids known in other literature. On the other hand, the reason why the bioavailability of flavonoid glycosides is low is that it is rapidly transformed in vivo, so it is low to calculate the bioavailability with the prototype. Therefore, the bioavailability evaluation of flavonoid glycosides should be combined with the comprehensive bioavailability evaluation of metabolites.

In the pharmacokinetic analysis, an LC-MS method for quantification was developed and validated, and applied in a pharmacokinetic study after oral administration of complanatuside in rats. From the results, we can find that complanatuside was rapidly converted into rhamnocitrin 3-*O*-β-glc and rhamnocitrin because of degradation of glucosides in vivo, which cause a low bioavailability of complanatuside. This method will be useful to explain the metabolic behaviors of similarflavonoids in the body and to investigate the crucial medicinal substance of complanatuside.

## 4. Materials and Methods

### 4.1. Chemicals and Reagents

Complanatuside, rhamnocitrin 3-*O*-β-glc, rhamnocitrin, and internal standard (IS, quercetin) with a purity of more than 98% were provided by Institute of Clinical Pharmacology, Guangzhou University of Chinese Medicine (Guangzhou, China). Their structures were confirmed using MS and ^1^H- and ^13^C-nuclear magnetic resonance (NMR) spectroscopy [27]. The chemical structures of complanatuside and quercetin are shown in Figure 6. Deionized water was purified using a Millipore water purification system (Millipore, Billerica, MA, USA). Acetonitrile and methanol used in the study were all UPLC-MS pure grade (Fisher Chemical Company, Geel, Belgium). Formic acid was purchased from the company of Sigma-Aldrich. Other reagents of analytical grade were purchased from Guangzhou Chemical Reagent Factory (Guangzhou, China).

Complanatuside was dissolved in water to form a solution for oral administration with a concentration of 7.2 mg/mL. Blank rat plasma was prepared by our research group.

### 4.2. Animal Experiments

One hundred and sixty male Sprague-Dawley rats (200–250 g) obtained from the Laboratory Animal Center of Guangzhou University of Chinese Medicine were used for plasma collection. Animals were bred in a breeding room with a temperature of 24 ± 2 °C, relative humidity of 60 ± 5%, and 12 h dark-light cycle. They were given tap water and fed normal food ad libitum. Animal welfare and experimental procedures were strictly in accordance with the Guide for the Care and Use of Laboratory Animals (US National Research Council, 1996) and the related ethics regulations of this University (NO.712052), the table as shown in the Appendix A.

### 4.3. Sample Preparation

#### 4.3.1. Metabolism Study

The whole blood (150 μL) was collected at 0, 5, 15, 30 min, 1, 1.5, 3, 6, and 9 h after oral administration (72 mg/kg) of complanatuside solution from the fossa orbitalis vein. Additionally, all the blood samples were centrifuged (3800 r/min) for 15 min at 4 °C to obtain plasma. Plasma samples were mixed (50 μL) and pipetted into the 1.5 mL polythene tubes and then followed by methanol (200 μL). The mixture was vortexed for 3 min and centrifuged (13,000 r/min) for 10 min, and the supernatant was transferred and evaporated at 37 °C under a stream of nitrogen. The residue was reconstituted in the mobile phase (50 μL) and centrifuged (13,000 r/min) for 10 min and 2 μL was used for analysis.

Blank urine and dose urine were collected by using metabolism cages for 12 h after administration. After following by methanol (300 μL), each urine sample (400 μL) was centrifuged (13,000 r/min) for 10 min to obtain the supernatant which was transferred and evaporated at 37 °C under a stream of nitrogen. The residue was reconstituted and centrifuged, and 2 μL was used for analysis.

Blank and dose stool powder were also collected at the same time. After extracting each dried powder stool sample (100 mg) with 1 mL methanol, the mixture was ultrasonicated for 30 min in an ice water bath, which was then centrifuged (13,000 rpm, 4 °C, 10 min) to obtain the clear supernatant. Similarly, the supernatant was evaporated to a dried residue and supplemented with 500 µL of methanol to be reconstituted and centrifuged (13,000 rpm) for 10 min.

In terms of obtaining bile samples, the rats were anesthetized by urethane (1.0 g/kg). After abdominal incision surgery, aplastic cannula was inserted into the bile duct by surgery to collect bile. The blank and drug bile samples were collected for 12 h from the start of oral administration, and the treatment of them was consistent with the urine samples.

#### 4.3.2. Pharmacokinetic Study

Whole blood samples (about 150 μL) were collected in heparinized polythene tubes at 0, 5, 10, 15, 45 min, 1, 1.5, 2, 3, 4, 6, 8, 10, and 12 h after oral administration (72 mg/kg) of complanatuside solution, and immediately centrifuged (3800 r/min) for 15 min at 4 °C to obtain plasma, and all samples were kept at −80 °C. Frozen rat plasma samples were thawed to room temperature prior to preparation. Plasma samples (50 μL) were pipetted into the 1.5 mL polythene tubes and then followed by methanol (200 μL). The mixture was vortexed for 3 min and centrifuged (13,000 r/min) for 10 min, and the supernatant was transferred and evaporated at 37 °C under a stream of nitrogen. The residue was reconstituted in the mobile phase (50 μL) and centrifuged (13,000 r/min) for 10 min and 2 μL was used for analysis.

### 4.4. Instruments and Experimental Conditions

#### 4.4.1. UHPLC–Q-TOF-MS Conditions

Chromatographic analysis was performed using an UHPLC system (Shimadzu, Kyoto, Japan) consisting of Shimadzu LC-30AD binary pump, a Model SIL-30SD autosampler, an online degasser (DGU-20A5R), and a temperature controller for columns (CTO-30A). An Agilent C18 column (3.0 × 50 mm, 2.7 μm, Agilent Technologies Inc., USA) was carried out for separation. The column temperature was maintained at 25 °C. The autosampler was set at 4 °C. The mobile phase consisted of (A) acetonitrile and (B) water containing 0.1% formic acid using a gradient elution of 20–70% A at 0–12 min. The flow rate was 0.4 mL/min, injection volume was 2 μL.

An AB SCIEX TripleTOF 5600+ mass spectrometer (AB SCIEX, Foster City, CA, USA) was connected to the UHPLC system through an electro-spray ionization (ESI) interface. The ion source can be operated in negative mode, the ion spray voltage (ISFV) was set to −4500 V (negative ion mode); the turbo spray temperature (TEM) 550 °C; nebulizer gas (Gas 1), 55 psi; heater gas (Gas 2), 55 psi; and declustering potential (DP) −100 V, the spectra covered the range from *m*/*z* 100~1200 Da.

All data collected were processed using Analyst Software^TM^ 2.2 (AB SCIEX, Foster City, CA, USA).

Post-acquisition analyses were performed using PeakView^TM^ version 2.1 software (AB Sciex, Framingham, MA, USA) and MasterView^TM^ version 1.0 software (AB Sciex, Framingham, MA, USA) which employs a list of potential biotransformation target compounds, incombination with the built-in no-target screening.

#### 4.4.2. LC-MS/MS Conditions

The analysis of pharmacokinetics was conducted using an LC-MS/MS system consisting of an Agilent 1260–6460 liquid chromatography instrument (Agilent, Agilent Technologies Inc., Palo Alto, CA, USA) equipped with a quaternary pump, a vacuum degasser, a thermo-stated column oven, and an autosampler (set at 4 °C), which were coupled to a triple quadrupole mass spectrometer. Separation was performed on an Agilent C18 column (3.0 × 50 mm, 2.7 μm) at 25 °C. The mobile phase consisted of (A) acetonitrile and (B) water containing 0.1% formic acid with a gradient elution program (20–30% A at 0–1 min; 30–50% A at 1–2 min; 50–60% A at 2–3 min; 60–70% A at 3–6 min). The flow rate was kept constant at 0.4 mL/min and the injection volume was 2 μL. The ion source was ESI- Agilent Jet S; the gas flow rate was set at 5 L/min; nebulizer 45 psi; sheath gas temperature 300 °C; sheath gas flow 11 L/min; capillary 3500 V; and nozzle voltage 500 V.

The MassHunter^TM^ Workstation (Agilent, Waldbronn, Germany) was used for data collection and acquisition.

### 4.5. Method Validation

Validation of the analytical method was assessed on specificity, linearity, sensitivity, precision, accuracy, recovery, matrix effect, and stability compliance under the Food and Drug Administration of the United States (USFDA) guidelines.

Blank plasma from six different rats with and without analytes, and IS were used to evaluate the specificity, which is an indicator of whether endogenous interference has occurred. The calibration curves were created for quantitative analysis. Regression was accomplished using a linear equation with a weighting factor of 1/x^2^. The lower limit of quantification (LLOQ) was defined as the lowest concentration point of the calibration curve (S/N > 10). For evaluation of intra- and inter-day accuracy and precision, three concentration QC samples were analyzed repeatedly in a single day and three consecutive days along with the calibration curve. The variations of intra- and inter-day accuracy and precision were expressed as the relative standard deviation (RSD). The absolute recoveries were evaluated by comparing the peak area of the complanatuside in spiking extracted samples with the corresponding spiking un-extracted samples. Matrix effects were calculated by match spiking post-extracted blank plasma samples with corresponding standard clean solutions at three concentrations. The stability test of QCs in rat plasma was assessed in their store environment, including at room temperature (25 ± 1 °C) for 24 h, at −80 °C for 1 month, and three freeze (−20 °C) to thaw (room temperature) cycles. Measurements were taken six times.

### 4.6. Data Analysis of the Pharmacokinetic Study

According to DAS pharmacokinetic software package (version 3.2.8, Chinese Pharmacological Association, Anhui, China), the non-compartmental model was suitable to describe the pharmacokinetic parameters after oral administration. The main pharmacokinetic parameters such as the maximum plasma concentration (C_max_), the time to reach maximum drug concentration (T_max_), the area under the plasma concentration-time curve (AUC), half-life (t_1/2_), and mean residence time (MRT) were calculated.

## 5. Conclusions

In this study, the metabolism and pharmacokinetic studies of complanatuside were performed for the first time. A metabolic investigation of complanatuside was carried out through UHPLC-Q-TOF-MS/MS. The 34 metabolites and metabolic pathways were all characterized. In the pharmacokinetic analysis, a HPLC-MS/MS quantitative method for the main metabolites—rhamnocitrin 3-*O*-glucoside and rhamnocitrin—was developed and validated. The method was then successfully applied to the pharmacokinetic study after oral administration of 72 mg/kg of complanatuside. The results showed that complanatuside exhibited mild oral absorption (T_max_ = 1.0 h), fast elimination (t_1/2_ = 0.51 h), and poor absolute bioavailability (AUC_(0-t)_ = 143.52 µg/L·h). Overall, complanatuside metabolized as apro-drugand underwent further metabolism, and metabolized intothe main metabolites rhamnocitrin 3-*O*-glucoside and rhamnocitrin derivers. These findings could provide data and reference for further research and applications of complanatuside.

## Figures and Tables

**Figure 1 molecules-24-00071-f001:**
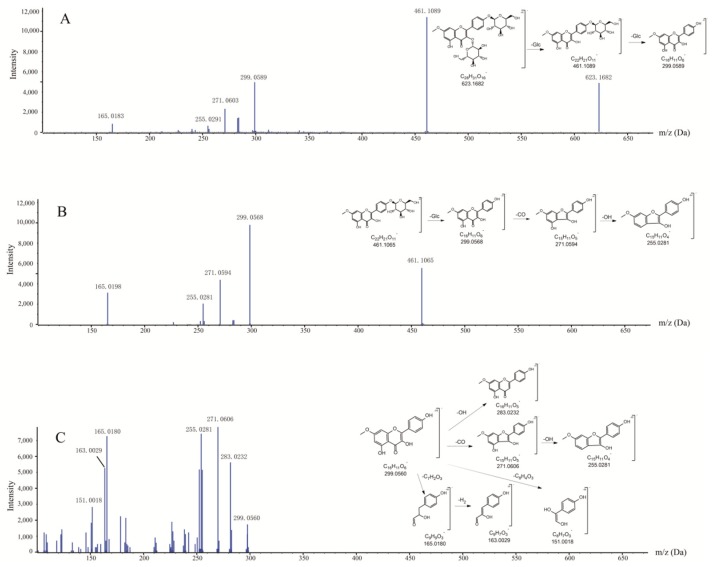
Proposed fragmentation pathways of complanatuside (**A**), RNG (**B**), and RNC (**C**). Abbreviation notes: RNG: rhamnocitrin 3-*O*-β-glc; RNC: rhamnocitrin.

**Figure 2 molecules-24-00071-f002:**
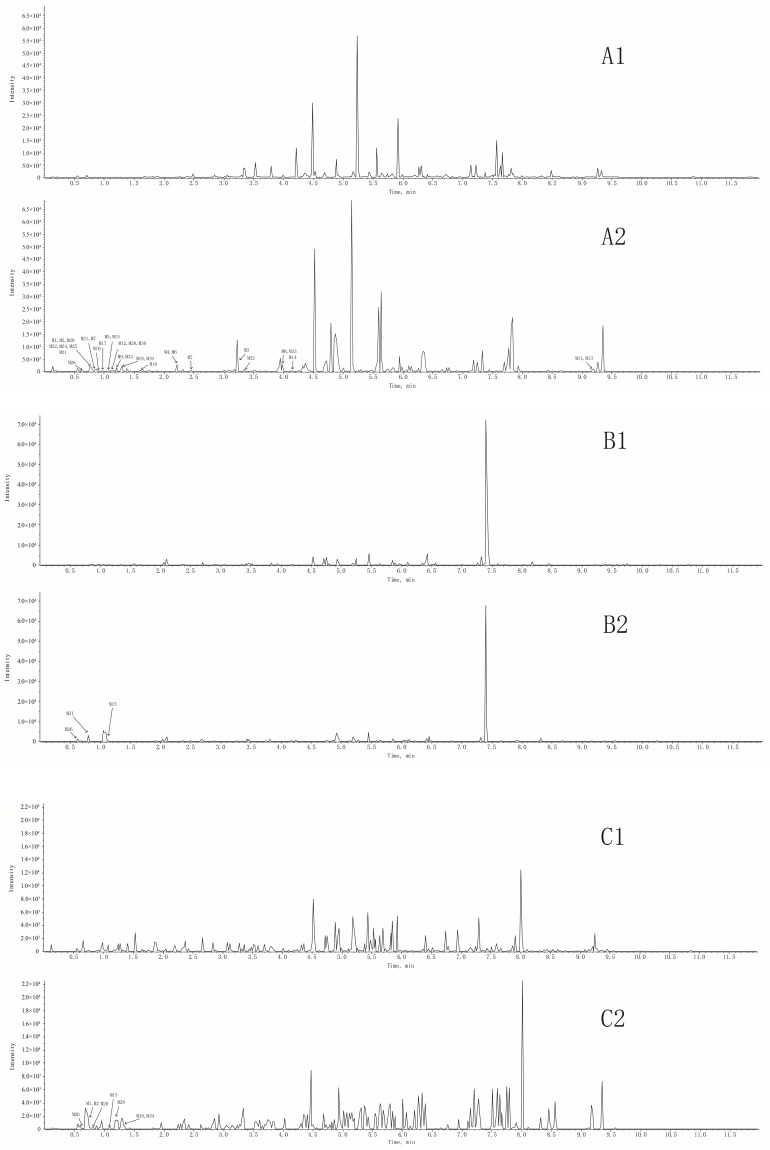
Total ion chromatograms of complanatuside in rat samples; (**A1**) blank plasma, (**A2**) plasma sample after oral administration; (**B1**) blank bile, (**B2**) bile sample after oral administration; (**C1**) blank stool, (**C2**) stool sample after oral administration; (**D1**) blank urine, (**D2**) urine sample after oral administration.

**Figure 3 molecules-24-00071-f003:**
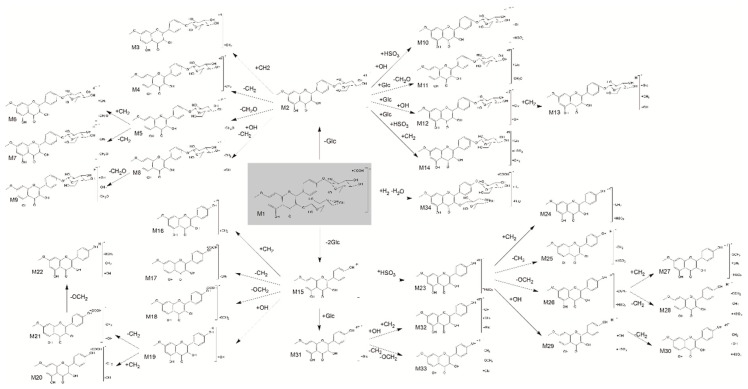
Proposed metabolic pathways of complanatuside.

**Figure 4 molecules-24-00071-f004:**
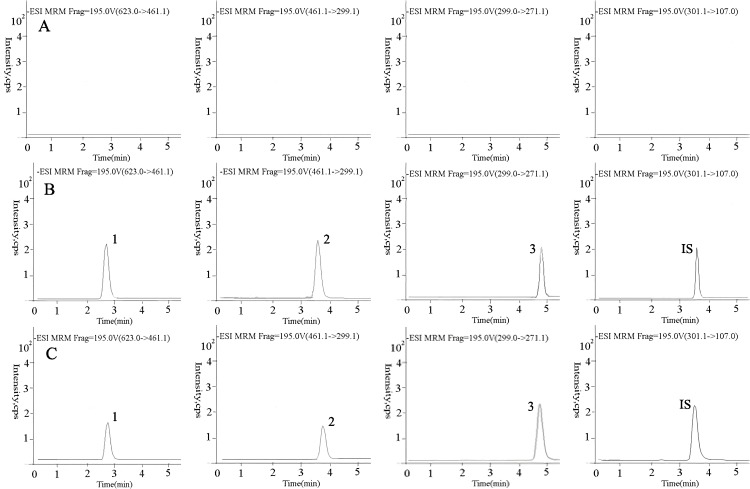
Representative multiple reaction monitoring (MRM) chromatograms of analytes in blank plasma samples (**A**); blank plasma samples spiked with the internal standard (**B**); the real plasma samples (**C**). (1) Complanatuside, (2) rhamnocitrin 3-*O*-β-glc, (3) rhamnocitrin, and (IS) quercetin. ESI: electro-spray ionization; IS:internal standard.

**Figure 5 molecules-24-00071-f005:**
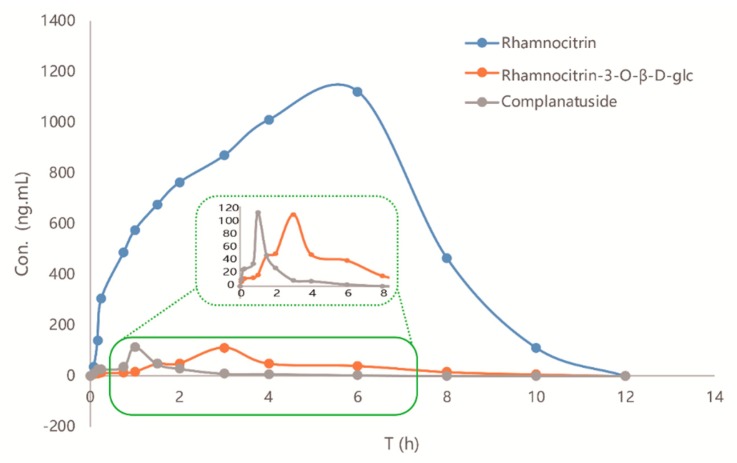
Time-concentration curves of CPS, RNG, and RNC in plasma from rats after oral administration of complanatuside. CPS: complanatuside; RNG: rhamnocitrin 3-*O*-β-glc; RNC: rhamnocitrin.

**Figure 6 molecules-24-00071-f006:**
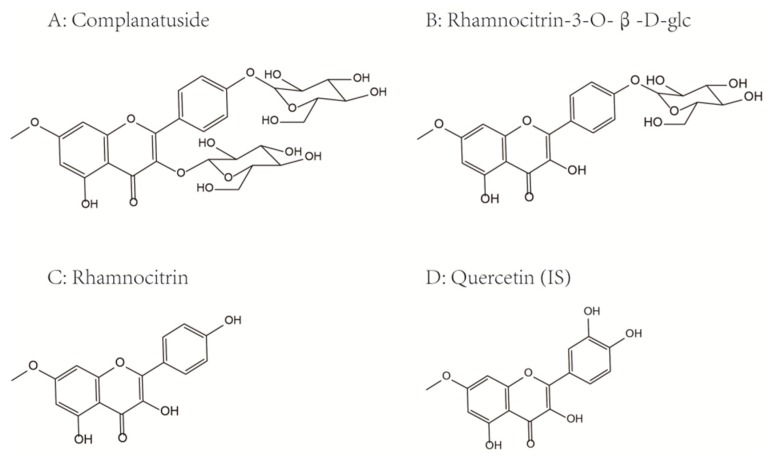
Chemical structures of complanatuside (**A**), rhamnocitrin 3-*O*-β-glc (**B**), rhamnocitrin (**C**), and quercetin (**D**, IS).

**Table 1 molecules-24-00071-t001:** Accurate mass measurements and metabolite description of complanatuside.

No.	Time (min)	Formula (M − H/+HCOO)	Experimental (Da)	Fragment Ion (Da)		Metabolite Description	Error (ppm)	Samples
**M1 ***	0.79	C_28_H_32_O_16_	669.1663	461.1089	299.0553	Parent	0.62	P, S *
**M2**	0.82	C_22_H_22_O_11_	461.1089	299.0558	283.0236	Deglycosylation	1.20	P, S
**M3**	3.20	C_23_H_24_O_11_	475.1246	429.2995		M2 methylation	1.18	P
**M4**	2.20	C_21_H_20_O_11_	447.0929	429.2978	149.0019	M2 demethylation	0.36	P
**M5**	1.04	C_21_H_20_O_10_	431.0983	299.1995	148.9996	M2 demethylation	1.11	P
**M6**	2.19	C_22_H_22_O_10_	445.1140	296.9965		M2 demethylation	1.18	P
**M7**	2.46	C_20_H_18_O_10_	463.0871	391.0617	152.9916	M2 demethylation	1.32	P
**M8**	3.99	C_21_H_20_O_12_	463.0882	431.1391	355.1693	M2 hydroxylation	1.18	P
**M9**	1.23	C_20_H_18_O_11_	479.0825	391.2848	227.2567	M2 hydroxylation	0.15	P
**M10**	1.58	C_22_H_22_O_15_S	557.0607	255.2311	181.9966	M2 sulfonation	1.05	P
**M11**	9.20	C_27_H_28_O_16_	653.1348	421.3522	175.0235	M2 glucuronidation	0.97	P
**M12**	1.14	C_28_H_30_O_18_	653.1359	447.1223	335.2226	M2 glucuronidation	0.78	P
**M13**	9.23	C_29_H_32_O_18_	713.1560	653.3056	447.1289	M2 glucuronidation	0.78	P
**M14**	4.12	C_29_H_32_O_20_S	731.1135	317.2127	299.1980	M2 glucuronidation	0.77	P
**M15**	1.08	C_16_H_12_O_6_	299.0559	285.0312	271.0606	Deglycosylation of M2	1.12	P, B *,S, U *
**M16**	0.92	C_17_H_14_O_6_	313.0718	285.1236	269.1281	M15 methylation	1.87	P
**M17**	1.09	C_15_H_10_O_6_	331.0460	269.1902	149.0965	M15 demethylation	2.12	P, U
**M18**	1.35	C_15_H_10_O_5_	269.0449	241.0491	225.0540	M15 demethoxylation	0.36	P, S, U
**M19**	1.28	C_16_H_12_O_7_	315.0510	300.0264	165.0183	M15 hydroxylation	1.65	P, S, U
**M20**	0.82	C_17_H_14_O_7_	375.0711	316.1675	285.1854	M15 hydroxylation	1.66	P, S, U
**M21**	0.91	C_15_H_10_O_7_	347.0398	285.2577	135.1503	M15 hydroxylation	1.68	P
**M22**	0.78	C_14_H_8_O_6_	271.0248	255.0763		M15 hydroxylation	2.01	P
**M23**	2.86	C_16_H_12_O_9_S	379.0129	299.0564	271.0616	M15 sulfonation	1.37	P
**M24**	0.76	C_17_H_14_O_9_S	393.0315	313.1804	299.1640	M15 sulfonation	1.39	P
**M25**	0.77	C_15_H_10_O_9_S	411.0017	287.1634		M15 sulfonation	1.49	P, U
**M26**	0.63	C_15_H_10_O_8_S	349.0024	331.2625		M15 sulfonation	1.56	P, B, S, U
**M27**	0.90	C_16_H_12_O_8_S	363.0180	285.0581	257.1902	M15 sulfonation	1.47	P
**M28**	1.17	C_14_H_8_O_8_S	334.9845	255.1022	240.0785	M15 sulfonation	4.95	P, S, U
**M29**	0.76	C_16_H_12_O_10_S	395.0078	347.2233	331.2281	M15 sulfonation	1.38	P
**M30**	1.17	C_15_H_10_O_10_S	380.9922	195.1384	181.1232	M15 sulfonation	1.43	P
**M31**	0.81	C_22_H_20_O_12_	475.0877	299.0553	284.0332	M15 glucuronidation	0.10	P, B
**M32**	3.36	C_23_H_22_O_13_	505.0988	447.1352		M15 glucuronidation	1.15	P
**M33**	4.07	C_20_H_16_O_11_	431.0620	355.1686		M15 glucuronidation	1.30	P
**M34**	1.21	C_28_H_32_O_15_	653.1761	447.1342	285.1258	M1 dehydration	7.11	P

* P: plasma; S: stool; B: bile; U: urine; **M1**–**M34**: 34 metabolites of complanatuside.

**Table 2 molecules-24-00071-t002:** The linear equation, correlation coefficients (*R*^2^), linear ranges, and lower limit of quantification (LLOQ) of CPS, RNG, and RNC in rat plasma.

Analytes	Linear Equation	*R* ^2^	Linear Range (ng/mL)	LLOQ (ng/mL)
CPS	Y = 0.0137X + 0.2118	0.9975	5.20–520.00	5.20
RNG	Y = 0.2714X − 0.4355	0.9951	2.04–510.00	2.04
RNC	Y = 0.2656X + 1.1438	0.9948	1.02–510.00	1.02

CPS: complanatuside; RNG: rhamnocitrin 3-*O*-β-glc; RNC: rhamnocitrin.

**Table 3 molecules-24-00071-t003:** The extraction recovery and matrix effect of CPS, RNG, and RNC in rat plasma (*n* = 6). RSD:relative standard deviation.

Analysts	QC Concentration (ng/mL)	Extraction Recovery		Matrix Effect	
		Accuracy (%)	RSD (%)	Accuracy (%)	RSD (%)
CPS	20	73.5 ± 8.7	5.5	102.1 ± 4.4	11.3
	125	81.0 ± 5.1	6.0	92.7 ± 14.6	8.3
	500	76.7 ± 4.3	6.4	98.7 ± 2.9	7.4
RNG	20	83.7 ± 2.8	12.6	86.0 ± 7.5	5.4
	125	83.3 ± 1.9	6.7	97.7 ± 2.1	5.9
	500	83.2 ± 5.1	6.1	98.2 ± 2.9	3.8
RNC	20	90.3 ± 3.7	7.5	101.1 ± 4.3	4.8
	125	89.0 ± 5.5	6.4	93.2 ± 2.3	3.3
	500	88.7 ± 3.7	9.8	95.6 ± 4.8	2.4

CPS: complanatuside; RNG: rhamnocitrin 3-*O*-β-glc; RNC: rhamnocitrin.

**Table 4 molecules-24-00071-t004:** The stability test of CPS, RNG, and RNC in rat plasma (*n* = 6).

Analytes	QC Concentration (ng/mL)	Post Preparation Stability		Short-Term Stability		Long-Term Stability		Freeze-Thaw Stability	
		Mean (ng/mL)	RSD (%)	Mean (ng/mL)	RSD (%)	Mean (ng/mL)	RSD (%)	Mean (ng/mL)	RSD (%)
CPS	20	18.6 ± 1.3	7.0	18.5 ± 1.4	7.6	17.4 ± 1.3	7.5	17.4 ± 1.2	6.9
	125	124.4 ± 11.2	9.0	124.1 ± 6.2	5.0	124.2 ± 9.1	7.3	122.1 ± 5.5	4.5
	500	489.3 ± 22.6	4.6	469.2 ± 12.5	2.7	486.5 ± 26.2	5.4	487.3 ± 23.7	4.9
RNG	20	19.3 ± 1.4	7.3	19.2 ± 1.3	6.8	18.7 ± 1.3	7.0	18.8 ± 0.9	4.8
	125	124.7 ± 10.9	8.7	123.6 ± 6.7	5.4	123.4 ± 10.6	8.6	123.5 ± 7.6	6.2
	500	491.5 ± 23.4	4.8	486.9 ± 23.3	4.8	479.4 ± 24.8	5.2	479.7 ± 12.8	2.7
RNC	20	19.2 ± 1.2	6.3	19.0 ± 1.1	5.8	19.7 ± 1.1	5.6	19.1 ± 0.8	4.1
	125	124.1 ± 10.5	8.5	124.0 ± 5.9	4.6	125.5 ± 6.9	5.5	124.7 ± 9.7	7.8
	500	476.3 ± 33.6	7.1	473.1 ± 25.1	5.3	495.3 ± 20.5	4.1	482.3 ± 14.3	3.1

CPS: complanatuside; RNG: rhamnocitrin 3-*O*-β-glc; RNC: rhamnocitrin.

**Table 5 molecules-24-00071-t005:** Intra-day and inter-day precision and accuracy of CPS, RNG, and RNC in rat plasma (*n* = 6).

Analytes	QC Concentration (ng/mL)	Intra-Day			Inter-Day		
		Actual Conc. (ng/mL)	Precision (RSD, %)	Accuracy (RE, %)	Actual Conc. (ng/mL)	Precision (RSD, %)	Accuracy (RE, %)
CPS	20	19.6 ± 0.6	3.1	2.0	19.8 ± 1.9	9.6	1.0
	125	124.8 ± 7.3	5.8	0.2	124.8 ± 6.2	5.0	0.2
	500	504.7 ± 24.2	4.8	−0.9	494.8 ± 17.9	3.6	1.0
RNG	20	19.2 ± 1.1	5.7	4.0	18.8 ± 1.7	9.0	6.0
	125	123.2 ± 5.8	4.7	1.4	125.4 ± 2.19	1.7	−0.3
	500	504.7 ± 21.4	4.2	−0.9	498.6 ± 18.2	3.7	0.3
RNC	20	19.3 ± 1.0	5.2	3.5	18.7 ± 1.9	10.1	6.5
	125	125.8 ± 4.9	3.9	−0.6	122.4 ± 10.5	8.6	2.1
	500	496.9 ± 13.6	2.7	0.6	505.8 ± 13.5	2.7	−1.2

CPS: complanatuside; RNG: rhamnocitrin 3-*O*-β-glc; RNC: rhamnocitrin.RE: relative error.

**Table 6 molecules-24-00071-t006:** The pharmacokinetic parameters of CPS, RNG, and RNC in rats after oral administration of complanatuside ($\bar{X}\pm S$, *n* = 8).

Analytes	C_max_ (ng/mL)	T_max_ (h)	t_1/2_ (h)	AUC_(0-t)_ (µg/L·h)	AUC_(0-∞)_ (µg/L·h)	MRT_(0-t)_ (h)
CPS	119.15 ± 11.25	1.00 ± 0.36	0.51 ± 0.04	143.52 ± 15.73	143.52 ± 15.73	1.72 ± 0.19
RNG	111.64 ± 14.68	3.00 ± 0.25	1.33 ± 0.55	381.73 ± 24.13	387.21 ± 28.06	4.28 ± 0.55
RNC	1122.18 ± 113.32	5.33 ± 0.63	4.15 ± 0.49	6540.14 ± 433.70	6627.61 ± 471.83	4.99 ± 0.11

CPS: complanatuside; RNG: rhamnocitrin 3-*O*-β-glc; RNC: rhamnocitrin; C_max_: the maximum plasma concentration; T**_max_**: the time to reach the maximum drug concentration; t_1/2_: half-life; AUC: the area under the plasma concentration-time curve; MRT: the mean residence time.

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
