# Peer review of "Identification and Pharmacokinetic Studies on Complanatuside and Its Major Metabolites in Rats by UHPLC-Q-TOF-MS/MS and LC-MS/MS"

_molecules, 2018, doi:10.3390/molecules24010071_

Round 1
Reviewer 1 Report
The paper entitled “Identification and Pharmacokinetic Studies on Complanatuside and its Major Metabolites in Rats by UHPLC-Q-TOF-MS/MS and LC-MS/MS” reports the metabolic and pharmacokinetic study on the complanatuside, used in Chinese medicine as a chemical marker for quality control of the drug. After reviewing, my comments concerning the manuscript are general positive, but before acceptance for publication in Molecules, the results need some modification considering following suggestions:
In order not to have problems with assigning the abbreviations, please, explain them under each table (for example compounds M1-M34, samples P,S,V in the Table 1; LLOQ in the Table 2; all pharmacokinetic parameters in the Table 6; QC concentrations in the Tables 3,4,5) . The reader, in this way, will be able to move more efficiently around the individual parts of the manuscript.
Based on literature, please expand the discussion. What direction of activity results from the metabolic pathway of the flavonoid polyphenols found in the complanatuside? Please discuss based on literature, what is the bioavailability of the investigated flavonoids and metabolites?
Please change in the Figure 6 (page 8), the sugar rings for the chair conformation.
Line 177, please correct “r2” (not r2).
The authors should also pay attention to the numbering of bibliography in the text. Please do not use the superscripts
I recommended that present manuscript is accepted, but after a very careful minor revision.
Author Response
Dear reviewer,
First of all, thank you for your time to review my manuscript.The following is my response to the review comments item by item.Attached is the revised manuscript, the modified parts are highlighted in blue.
The paper entitled “Identification and Pharmacokinetic Studies on Complanatuside and its Major Metabolites in Rats by UHPLC-Q-TOF-MS/MS and LC-MS/MS” reports the metabolic and pharmacokinetic study on the complanatuside, used in Chinese medicine as a chemical marker for quality control of the drug. After reviewing, my comments concerning the manuscript are general positive, but before acceptance for publication in Molecules, the results need some modification considering following suggestions:
In order not to have problems with assigning the abbreviations, please, explain them under each table (for example compounds M1-M34, samples P,S,V in the Table 1; LLOQ in the Table 2; all pharmacokinetic parameters in the Table 6; QC concentrations in the Tables 3,4,5) . The reader, in this way, will be able to move more efficiently around the individual parts of the manuscript.
1. Based on literature, please expand the discussion. What direction of activity results from the metabolic pathway of the flavonoid polyphenols found in the complanatuside? Please discuss based on literature, what is the bioavailability of the investigated flavonoids and metabolites?
[Answer] Thank you very much for your comments. As mentioned in the article, numerous studies have shown that flavonoids and its metabolites, such as flavonoid polyphenols, are capable of scavenging oxygen radicals, anti-inflammatory and other biological activities. At the same time, I also supplemented the recent research literature.
Well, the reason why the bioavailability of flavonoid glycosides is low is that it is rapidly transformed in vivo, so it is low to calculate the bioavailability with the prototype. Therefore, the bioavailability evaluation of flavonoid glycosides should be combined with the comprehensive bioavailability evaluation of metabolites. This part is also discussed in the discussion section.
2. Please change in the Figure 6 (page 8), the sugar rings for the chair conformation.
[Answer] Thanks for your instruction. We have revised as the suggestion.
3. Line 177, please correct “r2” (not r2).
[Answer] Thanks for your instruction. We have revised as the suggestion and correct it as “R2”.
4. The authors should also pay attention to the numbering of bibliography in the text. Please do not use the superscripts.
[Answer] Thanks for your careful review. We have checked and renumbered the bibliography in the manuscript, and also avoid using superscripts.
I recommended that present manuscript is accepted, but after a very careful minor revision.

Reviewer 2 Report
Although mentioned, in the paper there are not the five figures and the six tables quoted. Since the tables show the data on the validation of the method, the article cannot be totally evaluated. The evaluation is postponed when the tables and figures quoted are present in the paper.
Nevertheless, it has to be said that Authors should justify why the study was carried out in animal model and how data should be translated to human.
Author Response
Dear reviewer,
First of all, thank you for your time to review my manuscript.
My dear reviewer, I think there may be some misunderstanding. I uploaded the figures and tables as attachments. I thought that you could see them. This is our negligence, now, I insert the figures and tables into the text and convert it to PDF format for your review. Thank you very much.
Attached is the revised manuscript, the modified parts are highlighted in blue.

Round 2
Reviewer 2 Report
The authors fairly answer to reviewers questions . The article is now acceptable for publication.